# Glucomannan engineering highlights roles of galactosyl modification in fine-tuning cellulose-glucomannan interaction in Arabidopsis cell walls

Yoshihisa Yoshimi ®[1], Li Yu ®[1], Rosalie Cresswell[2], Xinyu Guo[1], Alberto Echevarría-Poza[1], Jan J. Lyczakowski ®[3], Ray Dupree ®[2], Toshihisa Kotake ®[4] & Paul Dupree ®[1] ✉

Widely found in most plant lineages, β-mannans are structurally diverse polysaccharides that can bind to cellulose fibrils to form the complex polysaccharide architecture of the cell wall. How changes in polysaccharide structure influence its cell wall solubility or promote appropriate interaction with cellulose fibrils is poorly understood. Glucomannan backbones acquire variable patterns of galactosyl substitutions, depending on plant developmental stage and species. Here, we show that fine-tuning of galactosyl modification on glucomannans is achieved by the differing acceptor recognition of mannan α-galactosyltransferases (MAGTs). Biochemical analysis and [13]C solid-state nuclear magnetic resonance spectroscopy of Arabidopsis with cell wall glucomannan engineered by MAGTs reveal that the degree of galactosylation strongly affects the interaction with cellulose. The findings indicate that plants tailor galactosyl modification on glucomannans for constructing an appropriate cell wall architecture, paving the way to convert properties of lignocellulosic biomass for better use.

Plant cell walls are a complex polysaccharide network that comprises 80% of the biomass of living organisms in the biosphere.[1] The versatile utility of plant cell wall polysaccharides has gained great attention to achieve a recycling-oriented society with a low carbon footprint, yet a basal understanding of the polymers is still limited. Together with industrial importance, plant cell walls are essential for plant growth and development as they give the cell rigidity to bear an osmotic pressure and define cell shape[2,3]. The functionality of the cell wall owes to the molecular architectural arrangement of a range of polysaccharides that are mainly composed of cellulose, hemicelluloses, and pectin. Hemicelluloses are a group of polysaccharides that can be extracted by alkali treatment and that share a β−1,4-linked backbone structure: xyloglucan, xylan, and β-mannan in eudicots[2,4]. It has been shown that hemicelluloses aid different mechanical properties of the bacterial cellulose-based hydrogel,[5] yet the relationship between the structure of hemicellulose and the interaction with cellulose, which is the foundation of the formation of the cell wall architecture, remains to be elucidated.

β-mannans, one of the major hemicelluloses ubiquitously found throughout the plant kingdom, can be classified into several subgroups based on their structural features. Homomannan purely consists of β−1,4-linked mannose (Man) whereas glucomannan is

[1]Department of Biochemistry, University of Cambridge, Hopkins Building, The Downing Site, Tennis Court Road, Cambridge CB2 1QW, UK. [2]Department of Physics, University of Warwick, Coventry CV4 7AL, UK. [3]Department of Plant Biotechnology, Faculty of Biochemistry, Biophysics and Biotechnology, Jagiellonian University, Gronostajowa 7, Krakow 30-387, Poland. [4]Division of Life Science, Graduate School of Science and Engineering, Saitama University, 255 Shimo-okubo, Sakura-ku, Saitama 338-8642, Japan. ✉e-mail: pd101@cam.ac.uk

composed of β−1,4-linked Man and glucose (Glc) with a random arrangement. The mannosyl residues of both polymers are often modified by α−1,6-linked galactosyl (Gal) side chains, making them galactomannan and galactoglucomannan, respectively. The galacto-glucomannan may also have acetyl groups at C2 and/or C3, and so it is conventionally categorised as acetylated galactoglucomannan (AcGGM). In eudicots, the AcGGM had been thought to be the only β-mannan in cell walls; however, a structurally distinctive β-galactoglucomannan (β-GGM) has recently been found widely in eudicot primary cell walls.[6] β-GGM has a Glc-Man repeating unit in the backbone, and a single α-Gal or β-Gal-(1→2)-α-Gal disaccharide on the mannosyl residues via the α-1,6-linkage. The structural diversity opens up questions of how the β-mannans with different structures are syn-thesised and behave differently in plant cell walls.

Among such considerable structural variation, α-1,6-Gal is a common and critical modification of β-mannans. We know that, in vitro, the Gal side chains keep the polymers soluble. As it has no substitution, homomannan aggregates forming a crystalline structure in a manner similar to chains of cellulose,[7,8] and similarly, the removal of Gal from galactomannan reduces its solubility, and it forms aggre-gates eventually.[9] Moreover, the Gal side chains also affect the inter-action of mannan polysaccharides with bacterial cellulose and other polysaccharides in vitro.[5,10–12] Although the fine structure and sub-stitution of β-mannans have clear effects on the polysaccharides' interactions with cellulose in vitro, the role in the plant cell wall is far from clear. The interaction of AcGGM/β-GGM with cellulose in plant cell walls has been observed by an advanced solid-state nuclear mag-netic resonance (ssNMR) technique with [13]C-labelled plants.[6,13,14] How-ever, the effect of α-1,6-Gal side chains on the molecular behaviour of AcGGM in plant cell walls and the interaction with cellulose remains obscure.

Plants synthesise β-mannans with diverse, yet appropriate, structures for the function in different cell wall contexts. This implies a precise control of the synthetic mechanisms; however, biosynthesis of diverse β-mannan is not fully understood. β-Mannan biosynthesis begins by a CELLULOSE SYNTHASE LIKE A (CSLA) in the glycosyl-transferase (GT) 2 family,[15] polymerising Man and Glc in the Golgi apparatus. Galactosylation is catalysed by enzymes called Mannan Alpha-Galactosyl Transferases (MAGTs) that belong to CAZY GT34 family together with xyloglucan xylosyltransferases (XXTs).[16–20] MAGT activity was first detected on homomannan from leguminous seeds.[16,21] After the discovery of the MAGT activity, the frequency of the Gal substitution of tobacco seed galactomannans was changed by over-expressing an MAGT from fenugreek.[22] Recently, we demonstrated AtMAGT1 has activity specifically towards the patterned [Glc-Man] disaccharide repeating structure from seed mucilage.[20] Nevertheless, no other MAGT activity has been experimentally shown, so it is unclear whether other MAGTs recognises a specific acceptor substrate or are promiscuously active. MAGT enzymes with different substrate speci-ficities would be advantageous for AcGGM engineering and be a compelling tool to address the effect of the Gal side chain frequency on AcGGM-cellulose interactions.

Here, we show that four MAGTs from representative plant species harbouring distinctive β-mannan structures differ in vitro in their recognition of backbone acceptor substrates. We achieve gluco-mannan engineering in Arabidopsis secondary cell walls with the MAGTs and a synthetic biology approach. Biochemical assays and solid-state magic-angle spinning (MAS) nuclear magnetic resonance (NMR) spectroscopy reveal that the behaviours of the polymers *in planta* differ depending on the degree of galactosylation. We propose that the presence of Gal promotes AcGGM binding to cellulose, yet an excess amount and clustered Gal side chains exhibit the opposite effect on the AcGGM-cellulose interactions. Our findings suggest that galactosylation of AcGGM is a well-controlled biosynthetic process in order to achieve the appropriate cell wall architecture.

## Results

### Substrate recognition of MAGTs contributes to the diversity of galactosyl modification on β-mannan

Plants in nature exhibit a variety of β-mannan structures where galactosyl residues are often seen on distinct backbone structures, i.e., homomannan, AcGGM, and β-GGM (Fig. 1a). We hypothesised that the diversity in galactosylation would partly be ascribed to differences of MAGTs in an acceptance of the backbone. For this purpose, we selected four MAGTs - *Arabidopsis thaliana* AtMAGT1 and AtMAGT2, *Pinus taeda* (pine) PtMAGT, and *Cyamopsis tetragonoloba* (guar bean) CtMAGT (Fig. 1b and Supplementary Fig. 1) - as representative plant species harbouring distinct β-mannan structures, β-GGM, AcGGM, and galactomannan, respectively. To perform in vitro activity assays on different β-mannan substrates, the selected MAGTs tagged with 3x-Myc were transiently expressed in *Nicotiana benthamiana* leaves and enriched in a microsomal membrane fraction (Supplementary Fig. 2b,c). PgGUX, a Golgi-localised glucuronosyltransferase for xylan from *Picea glauca*,[23] was similarly expressed as a negative control for any endogenous MAGT activity. In the assay, the expressed MAGTs were incubated with different acceptor polysaccharide substrates and then the polymeric fraction was digested by β-mannanase CjMan26A, β-mannosidase, and β-glucosidase to obtain only galactosylated oli-gosaccharides. The oligosaccharides obtained were separated and visualised using Polysaccharide Analysis by Carbohydrate gel Electro-phoresis (PACE). If no substitution arose through MAGT activity on acceptors, the tested polymer would be hydrolysed into mono-saccharides, whereas any galactosylated regions will appear as longer galactosylated oligosaccharides sensitive to α-galactosidase. We firstly examined their activities on homomannan prepared from ivory nut. As expected, CtMAGT produced galactosylated oligosaccharides (Fig. 1c), consistent with the previous report that guar beans synthesise galactomannan.[21] This activity was also confirmed with mannohexaose as an acceptor substrate (Supplementary Fig. 2d). The absence of activity from PgGUX excluded the possibility of any endogenous activity in tobacco microsomes. In contrast, none of the other MAGTs showed activity on homomannan, suggesting that they are specialised for galactosylation of glucomannan. To test the requirements of Glc residues in the acceptor substrate we used an alkali-extracted (and hence deacetylated) AcGGM from pinewood, from which Gal sub-stitution was also removed in advance by α-galactosidase. Indeed, AtMAGT1, AtMAGT2 and PtMAGT showed galactosylated products in the presence of the donor (Fig. 1d). The susceptibility of the products to α-galactosidase treatment confirmed the presence of galactosyl residues. Unexpectedly, CtMAGT also showed activity on AcGGM despite the lack of Glc in galactomannan from guar beans. These results indicate that there are types of MAGT that recognise only glu-comannan rather than homomannan, which is consistent with the previous observation with AtMAGT.[20]

Since we noticed subtle differences in the migration of oligo-saccharides in PtMAGT products compared with the others (Fig. 1d), we isolated the oligosaccharides from the gel for further analysis. As shown in the Supplementary Fig. 3, the sequential digestion with α-galactosidase, β-glucosidase, and β-mannosidase revealed that the major products were common in AtMAGT1, AtMAGT2, and CtMAGT as follows: 1, GAMM (where A is α-Gal-1,6-Man, M is Man, and G is Glc); 2, AMM or MAM; 3, AM. On the other hand, PtMAGT specifically pro-duced AGMM and GAM, where Gal was transferred to different posi-tions of Man relative to Glc. These results suggest that PtMAGT recognises Glc residues in the backbone at certain acceptor sites dif-ferently than the other MAGTs tested. To gain more insights into this specificity difference, structural models of MAGTs were generated by ColabFold[24] based on the XXT1 crystal structure.[25] The predicted MAGT models had average plDDT values of over 94, indicating high confidence level of the model structure (Supplementary Fig. 4). XXT1 has an acceptor cleft that is accommodated by cellohexaose (each

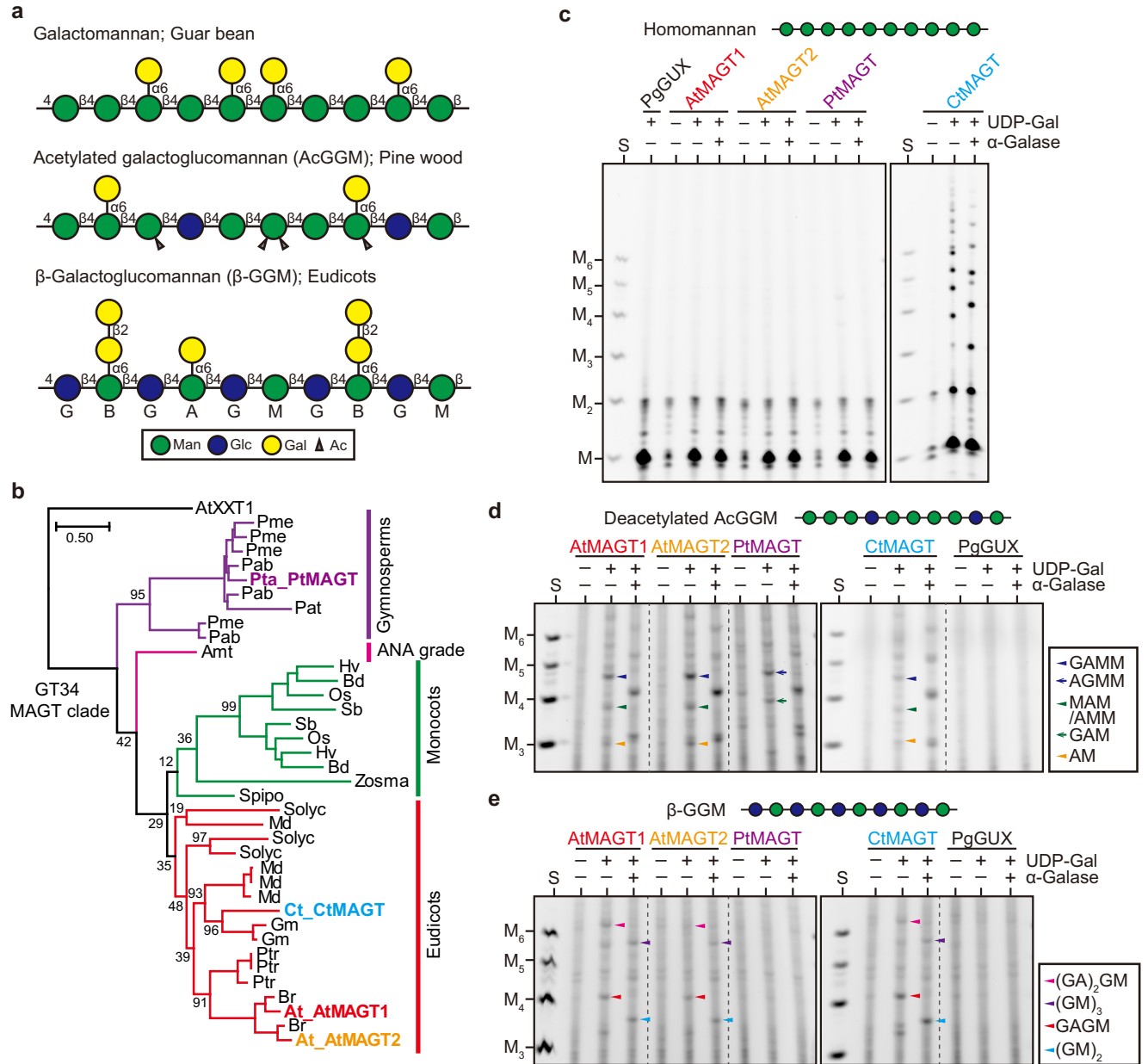

**Fig. 1 | MAGTs from different plant species display diverse acceptor recognition. a** Model structure of β-mannans modified with Gal side chains, which are found in different plant species. A single-letter nomenclature for the residues in the backbone and possible side chains are indicated. **b** An MAGT clade of GT34 phylogenetic tree. Amino acid sequences of MAGTs in different plant species were determined from the GT34 phylogenetic tree in Supplementary Fig. 1. AtXXT1 was selected as an outgroup. Highlighted MAGTs were characterised in the present study. Plant species were as followed: Pab, *Picea abies*; Pme, *Pseudotsuga menziesii*; Pta, *Pinus taeda*; Amt, *Amborella trichopoda*; At, *Arabidopsis thaliana*; Br, *Brassica rapa*; Ct, *Cyamopsis tetragonoloba*; Gm, *Glycine max*; Md, *Malus domestica*; Ptr,

*Populus trichocarpa*; Solyc, *Solanum lycopersicum*; Bd, *Brachypodium distachyon*; Hv, *Hordeum vulgare*; Os, *Oryza sativa*; Sb, *Sorghum bicolor*; Spipo, *Spirodela polyrhiza*; Zosma, *Zostera marina*. Scale bar represents 0.5 substitutions per site. In vitro activity of MAGTs towards homomannan (**c**), glucomannan (**d**), and patterned glucomannan (**e**). Two independent attempts yielded comparable results. The presence of Gal on the products was confirmed by α-galactosidase (α-Galase) treatment. S, standard of Man and mannooligosaccharides with D.P. 2-6. Structures of oligosaccharides in (**d**) were determined individually by further analysis shown in Supplementary Fig. 3. Source data are provided as a Source Data file.

monosaccharide occupies one subsite called S1 to S6, counting from the reducing end of the molecule), in which the transfer of xylosyl residue occurs at S4. At the S3 subsite, AtMAGT1, AtMAGT2, and CtMAGT have threonine (T239, T222, and T234, respectively) beneath the substrate, which could form a hydrogen bond with the axial C2-hydroxyl of Man (Supplementary Fig. 5). Intriguingly, the equivalent residue in PtMAGT was a leucine (L278), suggesting that PtMAGT prefers Glc to Man at the S3 subsite due to the steric obstacle against the axial C2-hydroxyl of Man. In fact, as PtMAGT produced AGMM in vitro, it is reasonable that PtMAGT preferentially recognises Glc at

the S3 subsite. It should be noted that PtMAGT also has the ability in vitro to produce GAM (Fig. 1d), suggesting that the S3 subsite could still accommodate mannosyl residues. Multiple alignments of MAGTs from various plant species showed that the leucine is conserved in gymnosperms, while most eudicots possess the threonine (Supplementary Fig. 5). Therefore, it is probable that the Glc recognition of this type of MAGTs is a common feature in gymnosperms.

To investigate whether acetylation of acceptor affects MAGT activity, a hot water-extracted AcGGM from pinewood was similarly tested for the in vitro activity assay. All MAGTs showed activity and

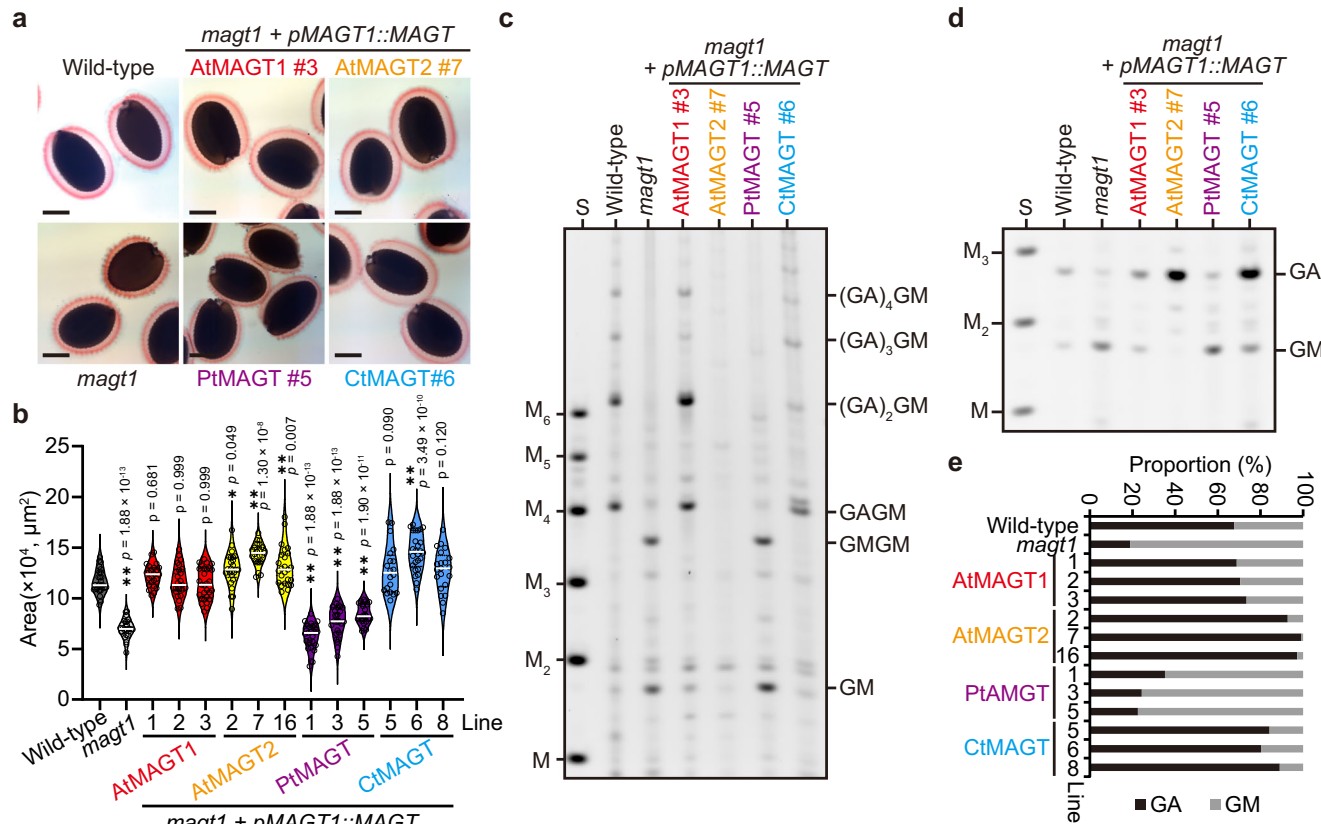

**Fig. 2 | In vivo activity of MAGTs towards patterned glucomannan.**
**a** Complementation of seed mucilage capsule of *magt1* mutant by four MAGTs. Bar = 100 μm. **b** Area of mucilage capsules. Open circles indicate individual measurements; the white lines represent the median of the group. One-way ANOVA (two-tailed) indicated a significant effect of genotype on mucilage area ($p = 1.87 \times 10^{-94}$; $F_{13,335} = 80.24$). Results of post hoc multiple comparisons (Dunnet's method compared with wild-type; wild-type, $n = 26$; *magt1*, $n = 25$; *AtMAGT1#1*, $n = 24$; *AtMAGT1#2*, $n = 25$; *AtMAGT1#3*, $n = 29$; *AtMAGT2#2*, $n = 25$; *AtMAGT2#7*, $n = 25$; *AtMAGT2#16*, $n = 25$; *PtMAGT#1*, $n = 27$; *PtMAGT#3*, $n = 27$; *PtMAGT#5*, $n = 24$; *CtMAGT#5*, $n = 20$; *CtMAG#6*, $n = 26$; *CtMAGT#8*, $n = 21$) are indicated by asterisks (*$p < 0.05$; **$p < 0.01$) with $p$ values. **c** CjMan26A-digestion profile of glucomannan

in seed mucilage of the complementation lines. Note that GMGM and GM were produced from PtMAGT seed mucilage, indicating that no galactosylation occurred. No product was detected in AtMAGT2 despite the recovery of the mucilage capsule. S, standards of Man and mannooligosaccharides with D.P. 2-6. **d** AnGH5-digestion profile of seed mucilage of complementation lines. AtMAGT2 produced almost only GA, indicating galactosylation occurred at almost all mannosyl residues. Two independent attempts yielded comparable results. Data from two independent biological replicates were provided in Supplementary Fig. 6. **e** Proportion of GA and GM determined from band intensity. Source data are provided as a Source Data file.

there was no difference in galactosylated products compared with those on the deacetylated glucomannan described above (Supplementary Fig. 2e), suggesting that MAGTs can transfer Gal residues on acetylated glucomannan. However, it was not evident whether MAGTs recognise mannosyl residues with acetyl groups or if they act on regions of this substrate with no acetylation.

For β-GGM biosynthesis, MAGTs need to act on the Glc-Man repeating backbone. To test MAGTs' capability of recognising such a structure, the β-GGM from seed mucilage was used for the assay. The activity of AtMAGT1, AtMAGT2 and CtMAGT on β-GGM was confirmed by detecting the oligosaccharides of GAGM and GAGAGM (Fig. 1e). Interestingly, no product was seen from the reaction with PtMAGT, revealing that the repeating pattern of β-GGM is not a suitable substrate for PtMAGT despite this enzyme's requirement for Glc in the acceptor.

To examine the differences in MAGT acceptor specificity further in vivo, we performed complementation of an Arabidopsis *magt1* mutant, which has a lack of Gal on β-GGM in seed mucilage as well as a reduced size of mucilage capsule.[19,20] The *MAGT* genes were introduced into the *magt1* mutant under the control of the *AtMAGT1* promoter (Supplementary Fig. 6a). Recovery of the defect in the mucilage capsule was confirmed in lines expressing AtMAGT1, AtMAGT2 and CtMAGT; however, PtMAGT did not restore the normal mucilage

phenotype (Fig. 2a-b; Supplementary Fig. 6). Furthermore, the mannanase CjMan26A digestion profile structure of glucomannan in seed mucilage from lines expressing AtMAGT1 and CtMAGT yielded an evenly spaced GAGM, GAGAGM, and longer, similar to wild-type seed mucilage[20,26] (Fig. 2c). In contrast, neither the mucilage capsule nor the glucomannan structure was recovered by the expression of PtMAGT. Therefore, together with the result of the in vitro activity assays, we conclude that the patterned glucomannan backbone of β-GGM is indeed an undesirable substrate for PtMAGT.

Curiously, no product was detected by mannanase digestion of the seed mucilage of *AtMAGT2* expressing lines, despite the recovery of the phenotype in the mucilage capsule. As the mannanase CjMan26A requires an unsubstituted mannosyl residue with no galactosyl modification at -1 subsite,[6,20] we hypothesised that all mannosyl residues of the patterned glucomannan in *AtMAGT2* mucilage are fully decorated with galactosyl residues. To investigate this, a mannanase from *Aspergillus nidulans* (AnGH5) that is tolerant to the Gal side chain at -1 subsite was used[6] (Supplementary Fig. 6). Indeed, almost only GA was liberated from the seed mucilage of *AtMAGT2* while the other MAGT expression lines showed a mixture of GA and GM (Fig. 2d,e). The relative ratio of GA/GM showed that the occurrence of galactosylation was more than 90% of mannosyl residues, indicating that AtMAGT2 is highly active on the mucilage glucomannan. The

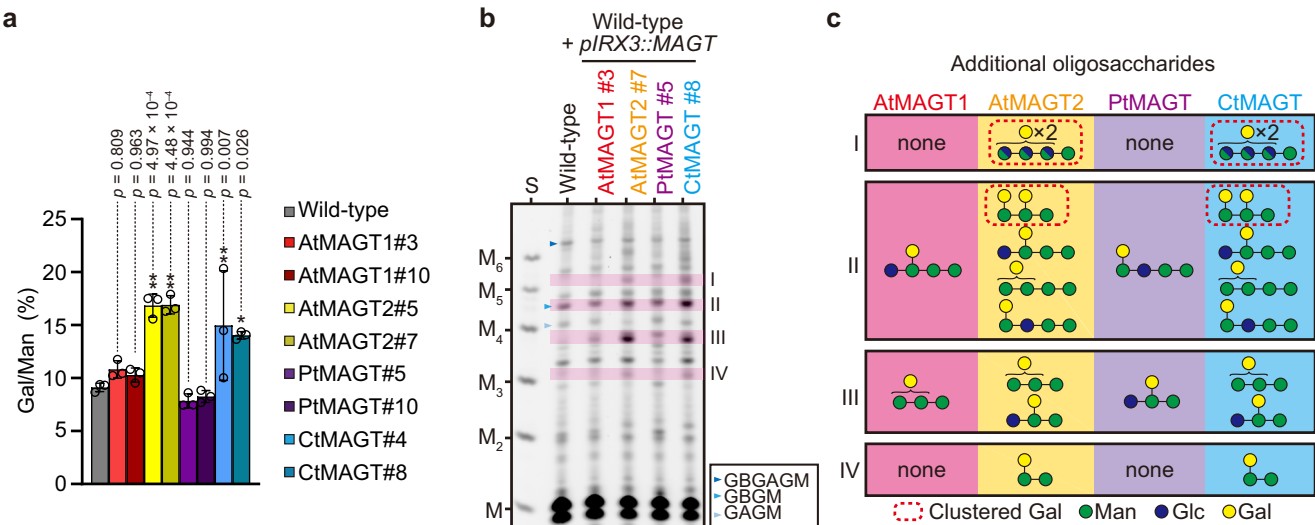

**Fig. 3 | Engineering of AcGGM in Arabidopsis secondary cell walls by MAGTs.**
**a** Gal/Man ratio of AcGGM in *pIRX3::MAGT* lines determined from the ratio of AnGH5-digested products. Data are mean values and standard deviation of three biological replicates. One-way ANOVA (two-tailed) indicated a significant effect of genotype on mucilage area ($p = 1.21 \times 10^{-5}$; $F_{8,18} = 11.41$). Results of post hoc multiple comparisons (Dunnet's method compared with wild-type) are indicated by asterisks (*$p < 0.05$; **$p < 0.01$) with $p$ values. Digestion profile and the measured proportion of products were shown in Supplementary Fig. 7. **b** Galactosylated oligosaccharide released from AcGGM in *pIRX3::MAGTs* expressing Arabidopsis. Unique bands found in *pIRX3::MAGTs* lines (I-IV) were further analysed (Supplementary Fig. 8). **c** Summary of galactosylated oligosaccharides found in the regions I-IV. Dotted squares indicate oligosaccharides with clustered galactosylation. Source data are provided as a Source Data file.

varying galactosylation level suggest that even the MAGTs active on the patterned β-GGM backbone exhibit different levels of activity. To investigate this difference, we considered the predicted structure of the MAGT acceptor binding sites, based on the homologous XXT enzymes in GT34. Culbertson *et al.* proposed that a xylosyl substitution could be accommodated in the S2 subsite the catalytic cleft of XXT5, a homologue of XXT1, due to its glycine residues at the position instead of isoleucine in XXT1.[25] Inspection of the different MAGT sequences revealed that the difference in MAGT activity may be explained by amino acid residues at the S2 subsite of AtMAGT2 where glycine G308 makes an additional pocket for Gal (Supplementary Fig. 5). In the other MAGTs, the equivalent residues were glutamate or aspartate which may sterically limit the binding of the acceptor with Gal substitution. The glycine at the S2 subsite was also found in some dicotyledonous species, such as *B. rapa*, *M. domestica*, and *S. lycopersicum* (Supplementary Fig. 5). Therefore, we propose that this type of MAGT is involved in making a different pattern of Gal modification on glucomannan.

Taken together, our in-depth characterisation of MAGT activities demonstrated that substrate recognition indeed varies depending on the source plant species and even between isozymes in a species. This suggests that the diverse β-mannan structures found in nature are consequences of the selective modification by MAGTs.

## AcGGM engineering in Arabidopsis secondary cell walls

To examine whether galactosylation of AcGGM can be fine-tuned *in planta* with the different substrate specificities of MAGTs, we introduced the *MAGT* genes under the promoter of *IRX3*, a secondary cell wall-specific promoter, into wild-type Arabidopsis (Supplementary Fig. 7). The bottom part of an inflorescence stem of Arabidopsis was used for biochemical analysis, of which AcGGM is the main glucomannan and it has a trivial amount of Gal.[6] Monosaccharide composition of total cell wall materials showed only a marginal or insignificant effect on the Gal amount in all MAGT expressing lines (Supplementary Fig. 7b), possibly due to the low abundance of AcGGM in Arabidopsis. However, the Gal/Man ratio of mannanase AnGH5 products significantly increased in *AtMAGT2* and *CtMAGT* lines (Fig. 3a;

Supplementary Fig. 7c, d). We then analysed AcGGM structures in detail by the combined digestion of CjMan26A mannanase, β-mannosidase, and β-glucosidase, after which only branched oligosaccharides can be detected. The mannanase-digestion profile exhibited great amounts of Man and Glc, indicating that majority of mannans degraded into the monosaccharides due to no branched structure; however, there were several products tolerant to the enzymatic digestion (Fig. 3b). Although we expected some branched oligosaccharides derived from β-GGM in primary cell walls, such as GAGM and GBGM[6] from the samples tested, all *pIRX3::MAGT* lines produced extra unique oligosaccharides in the region in the gel numbered as I-IV. The selected regions were excised from the preparative gels and oligosaccharides extracted for further analysis by sequential digestion with glycoside hydrolases (Supplementary Fig. 8). Figure 3c summarises the determined structures of galactosylated oligosaccharides found in the regions I-IV. Each *pIRX3::MAGT* line deposited AcGGM in secondary cell walls with a variety of galactosylation in accordance with the substrate specificity. For instance, *pIRX3::PtMAGT* lines displayed AGMM and GAM in regions II and III, respectively, whereas the other MAGTs preferably produced GAMM and/or AMMM/MAMM, and AMM/MAM. In addition, it was evident that AtMAGT2 and CtMAGT transferred Gal on AcGGM more frequently and more positionally variably. In particular, clustered galactosylation, such as the degree of polymerisation of (DP) 4 oligosaccharides with more than two Gal (region I) and consecutive galactosylation (AAM in region II), was observed. We noted that the proportion of oligosaccharides seemed to vary in AtMAGT2 and CtMAGT, which potentially resulted from their characteristic substrate specificities. Taken together, our results demonstrated that AcGGM structures can be modified with different galactosylation by taking advantage of the characteristics of MAGTs.

## α-galactosylation modulates AcGGM-cellulose interaction *in planta*

Having successfully engineered the decoration on AcGGM in Arabidopsis secondary cell walls, we could tackle the question how Gal side chains influence AcGGM-cellulose interactions. We hypothesised that the extractability of AcGGM may change if the interactions in the cell

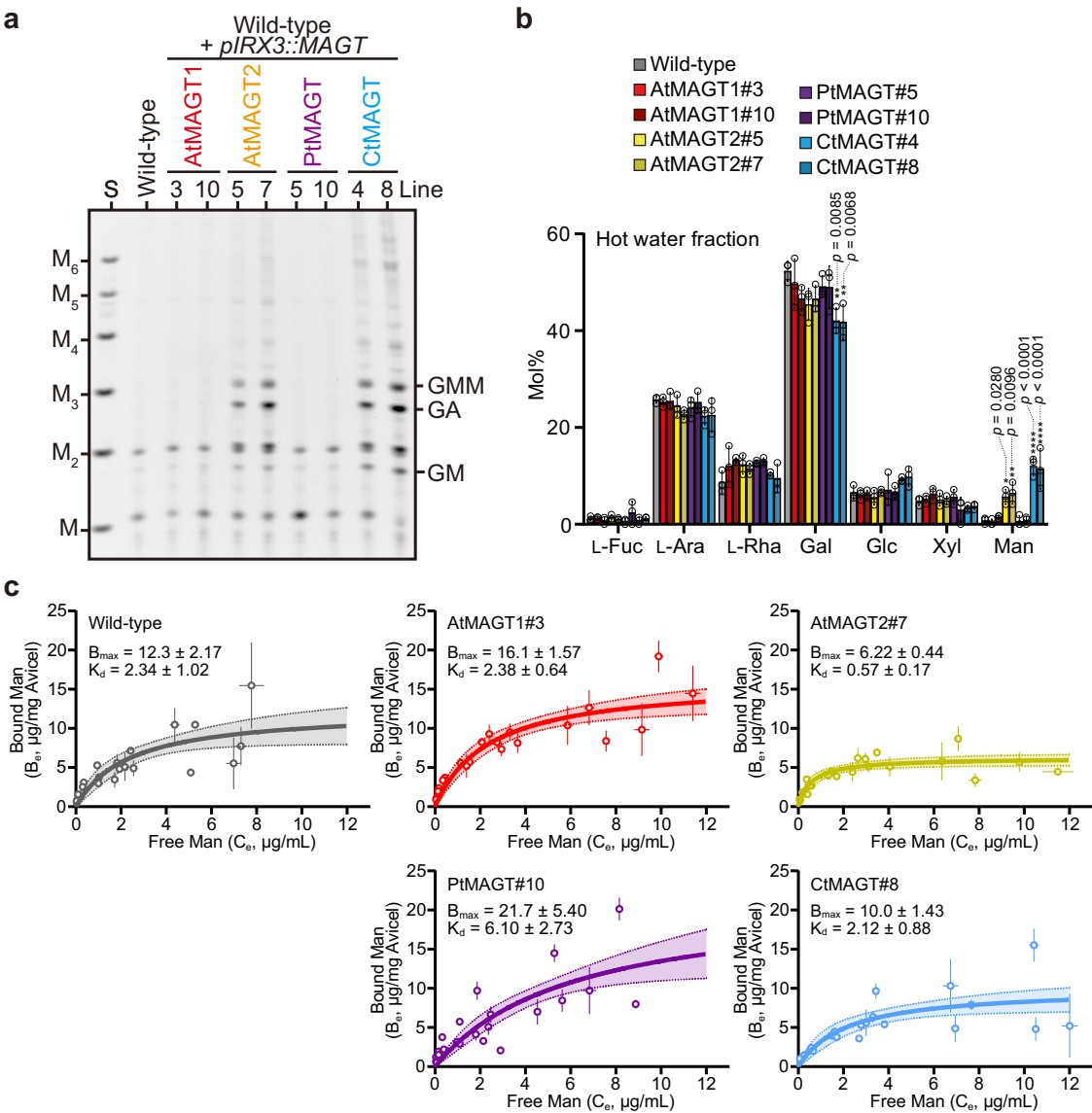

**Fig. 4 | Galactosylation changes the interaction of AcGGM with cellulose.**
**a** AnGH5-digestion profile of hot water extract from bottom stem AIR of *pIRX3::-MAGT* lines. The presence of products indicates the high extractability of AcGGM. S, standards of Man and mannooligosaccharides with D.P. 2-6. **b** Monosaccharides composition of the hot water extract. Data are mean values and standard deviation of three biological replicates. The significant difference was determined by one-way ANOVA on each monosaccharide, followed by Dunnets multiple comparisons where *pRIX3::MAGT* lines were compared with wild-type (*$p < 0.05$; **$p < 0.01$). *F* values and *P* values were provided in Supplementary Data 6. **c** In vitro cellulose-binding isotherms. KOH extracts of *pRIX3::MAGT* lines of a range of concentrations were incubated with and without Avicel. The amount of Man in the supernatant was determined by HPAEC-PAD, from which the bound Man was calculated. Data from three biological replicates were plotted in the graphs. Each point indicates a mean value with SEM of three technical replicates. Non-linear regression was used to fit the curves. The area with the shaded colour indicates the 95% confidence interval. $B_{max}$ and $K_d$ were calculated from the fitted curves and shown in the graphs with SEM. Source data are provided as a Source Data file.

walls are disrupted by the Gal modifications. To assess this idea, we performed hot water (HW) extraction on cell wall materials from the bottom part of the inflorescence stems. The mild heat treatment (80 °C for 4 h) extracted no AcGGM from wild-type (Fig. 4 and Supplementary Fig. 9), consistent with findings that alkali is needed for extraction,[27] since the innate AcGGM is tightly associated with cellulose. Remarkably, altered extractability of AcGGM in *pIRX3::AtMAGT2* and *pIRX3::CtMAGT* lines was evidenced by both the clear mannanase digestion profile and the increased proportion of Man in the HW fractions (Fig. 4). On the other hand, *pIRX3::AtMAGT1* and *pIRX3::Pt-MAGT* lines did not show any sign of the change in AcGGM extractability despite the presence of some galactosyl side chains (Fig. 3). It should be noted that the majority of AcGGM remained in the alkali-extractable fraction (87-93%) even in *pIRX3::AtMAGT2* and

*pIRX3::CtMAGT* lines (Supplementary Fig. 9), meaning that only a small portion of AcGGM is released in the relatively mild condition used in the extraction. These findings suggest that higher galactosylation weakens the AcGGM-cellulose interaction in the plant cell walls.

To test more directly whether galactosylation of AcGGM affects its interaction with cellulose, we performed in vitro cellulose adsorption assay with the alkali extracted-, deacetylated AcGGM from *pIRX3::MAGT* lines. By plotting the amount of bound Man at equilibrium versus free Man remaining in the solution, the maximum amount of Man binding to cellulose ($B_{max}$) and apparent $Kd$ can be measured. We decided not to take into account the amount of Glc for the isotherm as 1) other components, such as xyloglucan, contribute to the amount of Glc measured in the alkali extracts; 2) there is no major change in cell wall components except for Gal of AcGGM

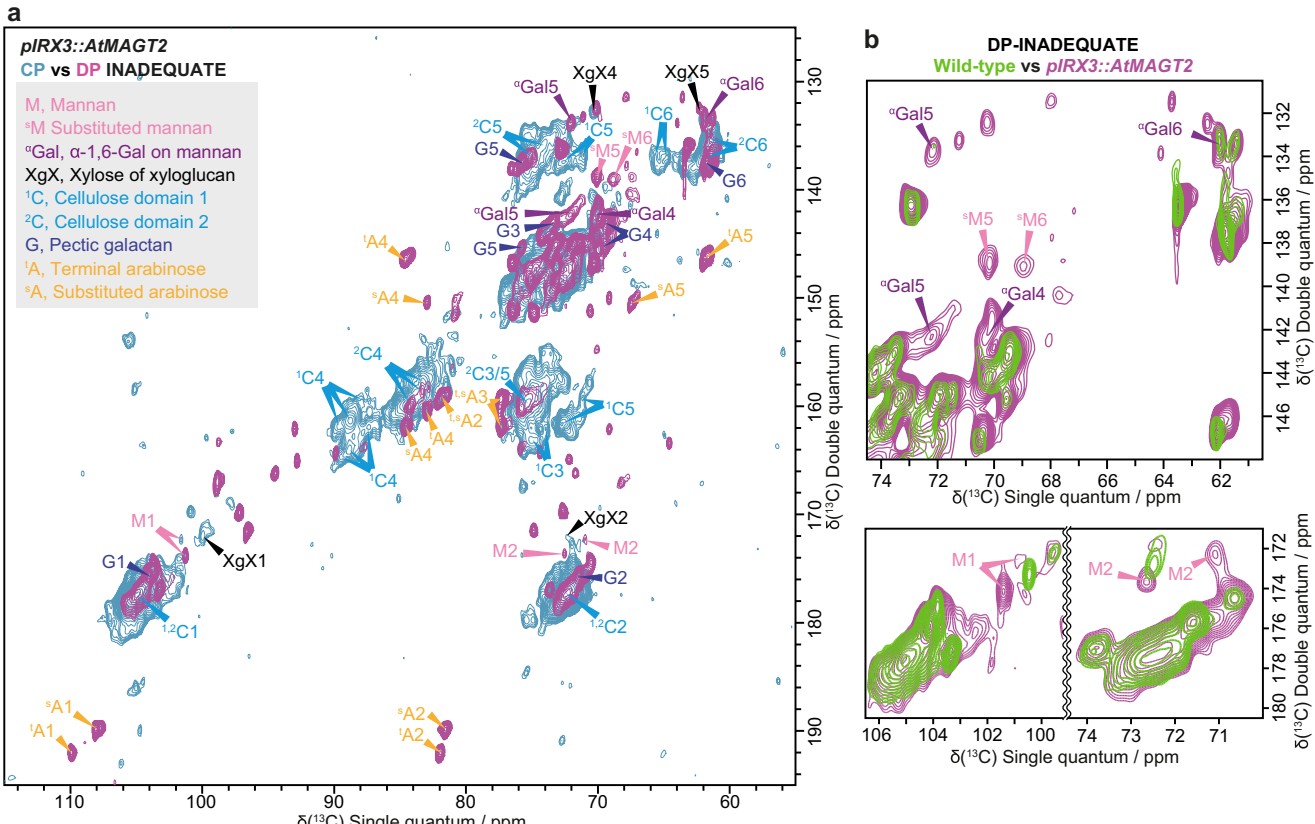

**Fig. 5 | Altered molecular behaviour of highly galactosylated AcGGM in secondary cell walls analysed by solid-state NMR. a** Overlay of [13]C CP- and DP-refocused INADEQUATE MAS ssNMR spectra of *pIRX3::AtMAGT2*. AcGGM peaks are labelled: Man (M), substituted Man (ˢM), α-Gal (Gal). Cellulose (domain 1, [1]C; domain 2, [2]C), xylose of xyloglucan (XgX), arabinose (A), and pectic galactan (G) are also labelled. Most AcGGM peaks were found in the DP spectrum, indicating its high mobility in the cell walls. **b** Comparison of C1, C2, and C5/6 regions of the DP spectra between wild-type and *pIRX3::AtMAGT2*. The spectra are normalised by C1 peaks at 105 ppm. Much less intensity of AcGGM peaks in the DP spectra of wild-type compared to *pIRX3::AtMAGT2*. Chemical shifts of the annotated peaks are listed in Supplementary Data 7. Spectra were acquired at a [13]C Larmor frequency of 213.8 MHz and a MAS frequency of 12.5 kHz. The spin-echo duration used was 2.24 ms.

(Supplementary Fig. 7b); and 3) the Glc/Man ratio was unchanged throughout the sample used here (Supplementary Fig. 6e). A clear reduction of the $B_{max}$ by half (6.22 µg/mg Avicel) was observed with the highly galactosylated AcGGM from *pIRX3::AtMAGT2* (Fig. 4), compared to the value of wild-type (12.3 µg/mg Avicel). There was a similar trend in *pIRX3::CtMAGT* (10.0 µg/mg Avicel). Interestingly, the opposite trends were seen in moderately galactosylated AcGGM from *pIRX3::AtMAGT1/PtMAGT* lines (16.1 and 21.7 µg/mg Avicel, respectively). These results indicate that the AcGGM-cellulose interaction is influenced by the galactosylation status of AcGGM.

To obtain further insight into the molecular behaviour of AcGGM and how galactosylation affects the interaction with cellulose *in muro*, we analysed [13]C-enriched, never-dried stems of *pIRX3::AtMAGT2* by two-dimensional (2D) magic angle spinning (MAS) ssNMR. A 2D INADEQUATE experiment where correlation peaks between two covalently bonded carbons are recorded, can deconvolute the complex cell wall matrix into individual carbons of different polysaccharides. Additionally, two experiments with cross-polarisation (CP) and direct-polarisation (DP) transfer are differently sensitive to the hydrodynamical state of polymers in cell walls.[6,14,28–30] The CP-MAS ssNMR mainly detects immobile components, such as cellulose and bound hemicelluloses, whereas DP-MAS ssNMR detects highly mobile cell wall components. Supplementary Fig. 10 shows a comparison of CP- and DP-refocused INADEQUATE MAS ssNMR spectra of wild-type stems. As expected, cellulose dominates the CP-INADEQUATE spectrum followed by hemicelluloses, such as xylan, xyloglucan, and AcGGM. The DP-INADEQUATE spectrum showed the peaks of mobile pectic galactan and arabinan, but no AcGGM-related peaks. For AcGGM, the peaks of Man and α-Gal side chain were seen in the CP spectrum and none of them were detectable in the DP spectrum, indicating the low mobility of AcGGM in the wild-type cell walls. It should be noted that there is a contribution of β-GGM in the primary cell walls, revealed by the presence of some peaks of substituted Man and α-Gal, which are unlikely AcGGM due to its minor Gal substitutions in Arabidopsis secondary cell walls. In contrast, a comparison of CP- and DP-INADEQUATE spectra of *pIRX3::AtMAGT2* in Fig. 5a exhibited enhanced AcGGM-related peaks in the DP spectrum. Especially, the peaks at 70.1 ppm and 68.9 ppm, which have previously been assigned as the substituted mannosyl carbon 5 (ˢM5) and ˢM6 of immobile β-GGM in Arabidopsis callus,[6] were significantly enhanced in the DP spectrum of *pIRX3::AtMAGT2* compared with those of wild-type (Fig. 5b). Likewise, the immobile AcGGM peaks were overall less in the CP-INADEQUATE spectrum of *pIRX3::AtMAGT2* compared to that of wild-type (Supplementary Fig. 10b).

These results support the notion that the high Gal substitution occurring on AcGGM makes it highly mobile in the cell walls, perhaps because of less binding of AcGGM to cellulose in the plant cell wall.

## Discussion

Unlike xylan and xyloglucan, which have simple backbone structures, β-mannans are unique polymers possessing structurally diverse backbones with differing proportions and arrangements of glucosyl residues amongst the mannosyl residues. The fact that plants reproducibly synthesise certain β-mannan structures opens up the question of how

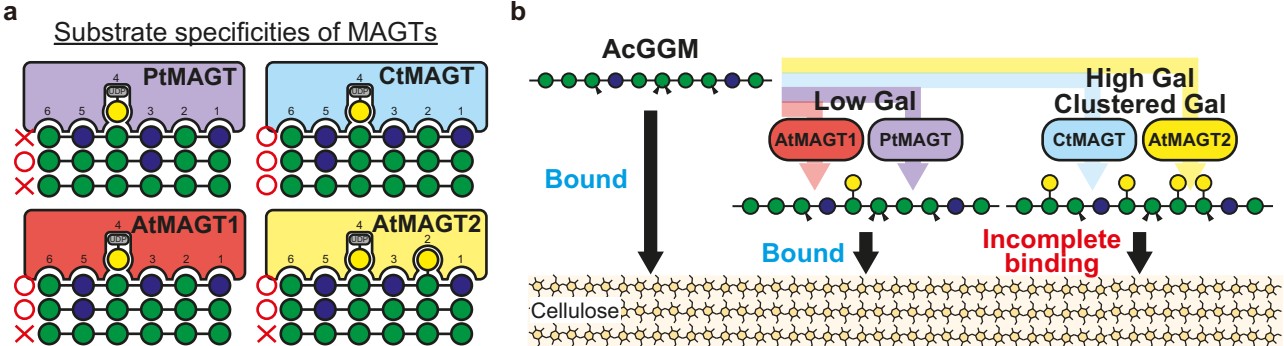

**Fig. 6 | Summary of substrate specificity of MAGT and effect of galactosylation on AcGGM-cellulose interaction. a** MAGTs characterised here showed different acceptor recognition. *Circles* indicate the substrates that MAGTs can recognise, while *Crosses* are substrates that MAGTs cannot act on. 1) AtMAGT1/2 and PtMAGT are specific to glucomannan while CtMAGT has a promiscuous activity towards any (gluco)mannan. 2) PtMAGT prefers Glc residues on the acceptor at subsite 3, and shows no activity on the patterned glucomannan. 3) AtMAGT2 has the additional pocket for Gal adducts, which allows it to transfer Gal near Gal branches on the acceptor. **b** Low galactosylation does not affect the interaction of AcGGM with cellulose *in muro* whereas high galactosylation ends up incomplete binding to cellulose presumably due to the loss of proper conformation to bind to cellulose and a steric hindrance of the high number of branches.

the galactosylation of β-mannan is controlled in vivo. Here, we showed that MAGTs have selective recognition of the acceptor substrates to define the appropriate modification of AcGGM. In the broader context of substrate specificity of GTs, the difference in the acceptor recognition has not been well investigated since hemicelluloses including xylan and xyloglucan have simple backbones. Here, our in-depth characterisation of MAGTs in vivo and in vitro provides new insight into how the substrate recognition of MAGTs differs between plant species and even between isozymes within a species.

As summarised in Fig. 6a, we revealed a specialisation of MAGTs towards AcGGM in Arabidopsis and pine. This explains why MAGT activity of PtGT34A (called PtMAGT here) from *P. taeda* was not detected previously where mannohexaose was used as the acceptor substrate.[31] This constrains the AcGGM biosynthesis in both plants to galactosylate the backbones where at least one Glc lies within the six residues that fit in the acceptor pocket of MAGTs. In particular, PtMAGT preferentially accommodates Glc at the S3 subsite likely due to the steric effect of non-polar Leu residue. The non-polar amino acid residue at S3 subsite is conserved across gymnosperms and is also found the one fern GT34 that belongs to the MAGT clade, suggesting that similar galactosylation could occur in ferns. However, it should be noted that most GT34s from ferns do not belong either to the XXT or to the MAGT clades in the phylogenetic tree, so further experimental investigation is needed to identify MAGTs in ferns. Since PtMAGT seeks Glc to act, the amount of Gal reflects the amount of backbone Glc. Indeed, AcGGM of softwood cell walls consists of the Man:Glc:Gal ratio of 4:1-2:0.3-1.2,[32–35] where the Gal amounts are similar to or less of that of Glc. It is worth noting that the structures of AcGGM in softwood cell walls seem variable in different extraction fractions.[34] Furthermore, the ssNMR experiments revealed that a portion of AcGGM irreversibly changed in mobility and the distance to cellulose upon a drying-rehydration cycle.[14] Thus, it is reasonable that there is a fine regulation on AcGGM biosynthesis, where the constraint on PtMAGT activity likely contributes to the structure found in pine cell walls. We did not test a homologous MAGT in *P. taeda*, which may also contribute differently to form variable AcGGM structures in pine cell walls.

Leguminous plants including guar beans are known to accumulate highly galactosylated galactomannans in their seed as a storage cell wall polysaccharide.[10,36] The promiscuous activity of CtMAGT towards glucomannans might not be relevant to the galactomannan biosynthesis in guar beans because the available substrates in the developing beans are homomannans. However, the relaxed substrate recognition might result from a structural adaptation of MAGT to enable guar beans to synthesise a highly substituted galactomannan.

Recently, it has been shown that two GT2 family enzymes, AtCSLA2 and AtCSLA9 in Arabidopsis, are responsible for the biosynthesis of β-GGM and AcGGM, respectively.[6] Although AtMAGT1 activity was previously determined for β-GGM biosynthesis,[20] this work revealed that it can accept any glucomannans regardless of the arrangement of Glc in the backbone as the substrate in vitro and in vivo, indicating that AtMAGT1 could galactosylate AcGGM as well as β-GGM. In the primary cell wall-rich tissues, both AcGGM and β-GGM coexist; however, AcGGM in wild-type Arabidopsis has little galactose modification.[6,37] This discrepancy implies that there is an unknown mechanism(s) controlling galactosylation of AcGGM and β-GGM, for instance, separation of biosynthetic machinery in time or space for AcGGM from that of β-GGM to limit the access of AtMAGT1 to the AcGGM backbone synthesised by AtCSLA9. This could be achieved by forming a biosynthetic complex of MAGT1 with AtCSLA2, compartmentalisation within the Golgi apparatus, or biosynthesis in different Golgi stacks of different cells. Such higher-order multiprotein complexes have been proposed for the evolutionarily related xyloglucan biosynthesis.[38–41] Further investigation is required to address mechanisms that limit MAGT activity towards β-GGM acceptor in the presence of AcGGM.

The functionality and property of the cell walls rely on the interaction between cellulose and hemicelluloses, in which structure and conformation of hemicelluloses play a critical role in maintaining the functional interactions. It has been shown that appropriate modification of xylan allows and stabilises the formation of the twofold screw conformation of the polymer which is important for the interaction with cellulose.[30,42,43] Thus, understanding how a specific modification contributes to the conformation of polymers and to the interaction with cellulose is essential. The molecular dynamics (MD) simulation has demonstrated that replacing Man in the mannooligosaccharides with Glc greatly increases the probability of twofold screw conformation due to the hydrogen-bond between 2-OH of Glc and 6-OH of Man, which strengthens the binding cellulose.[34,44,45] Interestingly, the Gal side chain also has a similar effect on the backbone conformation, and this effect seems enhanced when the Gal substitution occurs at reducing end side of Glc by a hydrogen-bond between the equatorial 2-OH group of Glc and Gal.[45] Similar effects have been seen in the previous hydrothermal extraction of AcGGM from spruce, which showed that recalcitrant AcGGM has a higher Gal level together with higher Glc.[34] Here, engineering of AcGGM modification was achieved by expressing MAGTs in Arabidopsis secondary cell walls, where we added Gal side chains to different degrees, reflecting the substrate recognition of MAGTs. In our MAGT-expressing system, the unchanged ratio of Glc to Man in the backbone disregards the effect of Glc on the interaction

with cellulose, so that we can consider the effect of Gal modification exclusively. In *pIRX3::AtMAGT1/PtMAGT*, there was no change in the AcGGM extractability despite the extra Gal modification compared with wild-type. Besides, the in vitro adsorption assay with AtMAGT1 and PtMAGT showed a trend of the increase in AcGGM binding to cellulose. These results might reflect that AcGGM has the conformation favouring the interaction with cellulose, i.e., twofold screw like conformation. We hypothesise that glucomannan-specific MAGTs have the refined substrate specificity in order to generate AcGGM structures suitable for binding to cellulose.

In the *pIRX3::AtMAGT2/CtMAGT*, on the other hand, the high galactosylation (15-20% of Man) led to the enhanced extractability of AcGGM as well as the reduction in the amount of bound Man to cellulose in the adsorption assay (Fig. 4). This was also supported by ssNMR where the high mobility of AcGGM peaks in the intact cell walls was evidenced (Fig. 5). Why does the increased Gal on AcGGM in *pIR-X3::AtMAGT2/CtMAGT* diminish the cellulose interaction and induce high mobility in the cell walls? Interestingly, it has been proposed that β-GGM in the primary cell walls interacts with cellulose strongly, even though approximately 50% of Man gets galactosylation.[6,20] This can be explained by the structural feature of β-GGM; in turn, the alternating units Glc-Man makes β-GGM have the Gal side chain on one side of the backbone with the even spacing. However, such a structural patterning was not detected in AcGGM of *pIRX3::AtMAGT2/CtMAGT*. They rather showed the presence of the consecutive galactosylation (denoted as AAM, Fig. 3), of which Gal branches locate towards both sides of the backbone, making AcGGM sterically hindered in certain interactions with cellulose. It has been shown in the MD simulation that consecutive Gal enforced the one Man to have a higher probability of gt and tg conformations.[45] Although we could not determine the conformational changes at carbon 6 of Man in ssNMR, the changed chemical shift at carbon 1 of Man could potentially be related to the conformational changes that occurred in high galactosylated AcGGM (Fig. 5).

Taken together, our results highlight how a relatively minor structural modification of glucomannan greatly affects the interaction with cellulose in plant cell walls, providing fundamental knowledge of the role of Gal substitution on the AcGGM-cellulose interaction (Fig. 6b). The diverse galactosylation of glucomannan observed in nature, which is driven by the different substrate specificities of MAGTs, could arise from an adaptive advantage of controlling AcGGM-cellulose interaction. However, the detailed mechanism of how AcGGM interacts with cellulose fibril surfaces requires further investigation. Furthermore, we should note that the acetylation of AcGGM would also affect the interaction with cellulose.[46,47] Although it is unclear whether the expression of them in Arabidopsis changed the degree of acetylation of AcGGM in vivo, MAGTs tested here were active in vitro on acetylated AcGGM from pine. For a better understanding of the AcGGM-cellulose interaction, it is vital to investigate further how two different modifications are coordinated in the biosynthetic machinery and the effect of acetylation on the cellulose interaction.

With a synthetic biology approach, we achieved glucomannan engineering *in planta*. The results demonstrate that a better understanding of polysaccharide biosynthesis provides a new technology to manipulate cell wall architecture. It may be possible to make large amounts of specifically structured GGM, using a suitable host that can synthesise large amounts of this polysaccharide, such as tomato fruits. The engineering of cell wall polysaccharides has a great potential to tailor the properties of the natural materials. For example, an ectopic deposition of callose in the plant cell walls has been shown to alter the hydration properties of woody biomass by making the cell walls more porous.[48] AcGGMs, such as *Amorphophallus konjac K.*, form a gel in vitro by deacetylation;[49,50] therefore, accumulation and structural modification of AcGGM in cell walls could be an alternative way to tailor the hydration properties of lignocellulose biomass. Given the

increasing values of β-mannans in the food industry and human health,[49,51–55] glucomannan engineering benefits not only for manipulating the properties as food additives, but also for increasing yield due to easier extraction without chemicals. Our work aids new fundamental insights on how Gal modification affects AcGGM-cellulose interactions, paving the way to design β-mannan structures for better industrial applications including foods, paper, textiles, and construction using wood.

## Methods

### Materials
Col-0 ecotype of Arabidopsis was used in all experiments performed here. A homozygous mutant of *magt1-1* (SALK_061576) was used.[20] Arabidopsis seeds were sterilised and sown on a half-strength of Murashige-Skoog-agar plate. After stratification at 4 °C for 2 days, seeds were germinated and grown at 21 °C under white light (MASTER TL-D Super 80 58 W/840 1SL/25 [Philips] and Sylvania 58 W T8 5 ft Grolux Tube [Sylvania]; 150 mmol $m^{-2}$ $s^{-1}$) with a 16-h light/8-h dark cycle. Two-week-old plants were then transferred to soil (Advance M2, ICL Levington) and grown in the same conditions. For ssNMR, two-week-old Arabidopsis plants were transferred onto rock wool and grown in a custom-built $^{13}CO_2$ chamber with a hydroponic solution.[30] After 6 weeks, the bottom part (typically less than 5 cm from the base) of the inflorescence stem was snap-frozen by liquid $N_2$ and stored at -80 °C until use. *Nicotiana benthamiana* tobacco was grown in the same conditions as Arabidopsis. Five- to six-week-old plants were used for the transient expression. Pinewood (*Pinus taeda*) for AIR preparation was obtained from the University of Cambridge Botanical Garden. Ivory nut homomannan (P-MANIV), cellohexaose (O-CHE), mannohexaose (O-MHE), and *Cellvibrio japonicus* Man26A β-mannanase (CjMan26A, E-BMACJ) were purchased from Megazyme. *Aspergillus nidulans* GH5 β-mannanase (AnGH5) and *Aspergillus niger* GH3 β-glucosidase were provided by Novozyme, *Cellvibrio mixtus* GH5 β-mannosidase was gifted from Harry Gilbert (University of Newcastle), and *Cellvibrio mixtus* GH27 α-galactosidase was from Prozomix. All purchased chemicals used in this work are listed in Supplementary Data 1.

### Phylogenetic analysis
The whole amino acid sequences of GT34 enzymes from various plant species were obtained from NCBI protein database (https://www.ncbi.nlm.nih.gov/protein/) and PLAZA database (https://bioinformatics.psb.ugent.be/plaza/) using Arabidopsis MAGT1 as a query. All sequences are listed in the Supplementary Data 2. The sequences longer than 600 amino acids and shorter than 280 amino acids were removed. All procedures to generate the phylogenetic tree were done by MEGA X software (version 10.2.6).[56] The sequences were aligned using the MUSCLE algorithm. The best model (WAG + G + I) was determined by a built-in tool in MEGA X. The phylogenetic tree was generated by the maximum likelihood method with a bootstrap value of 1000. Similarly, the phylogeny of the MAGT clade was constructed by the maximum likelihood method with the calculated model of JTT + G. Protein sequences of MAGT were selected from the phylogeny of the GT34 family. Multiple alignments were visualised by ESPript 3.0 web (https://espript.ibcp.fr).[57] Accession codes for the amino acid sequences of AtMAGT1, AtMAGT2, PtMAGT, and CtMAGT are At2g22900, At4g37690, DAA64590.1, and Q564G7.1, respectively.

### Prediction and comparison of MAGT protein structures
For the prediction of MAGT protein structures, only catalytic domains of MAGTs were used. To determine an N-terminal transmembrane domain of MAGTs, amino acid sequences were first subjected to DeepTMHMM ver 1.0.[58] The amino acid sequences of the catalytic domain were then subjected to ColabFold[24] for the prediction of individual proteins and their homodimers. The open-sourced Pymol

(version 2.5) was used for the visualisation and comparison of the predicted proteins. The crystal structure of AtXXT1 and its ligands, UDP and cellohexaose (PBD: 6bsw),[25] was also used for the comparison.

## DNA construct

Synthetic DNAs cording CDS of MAGTs obtained from GENEWIZ were fused with 3xMyc tag (pICSL50010; Addgene #50310) by Golden Gate cloning.[59] For the transient expression in *N. benthamiana*, the Myc-tagged MAGT sequences were amplified by PCR with Q5 DNA polymerase (Cat no. M0491, NEB) and primers listed in Supplementary Data 3, and ligated at *Nru*I site of a pEAQ-HT vector by T4 DNA ligase (Cat no. M0202, NEB).[60]

For the complementation of *magt1-1* mutant, the Myc-tagged *MAGT* genes were driven under the promoter of *AtMAGT1*, which was amplified by PCR from Arabidopsis genomic DNA with the primer set, pMAGT1_GG_F and pMAGT1_GG_R. For the modification of secondary cell wall AcGGM, Myc-tagged *MAGT* genes were driven by the secondary cell wall specific promoter, *IRX3*, which was obtained from GENEWIZ. By Golden Gate cloning,[59] these DNA fragments were first assembled into pICH47811 (Addgene #48008) as the level 1 part, and then cloned into a binary vector, pAGM4723 (Addgene #48015) at position 2. For the selection marker, a *pOleosin:Oleosin:eGFP:ActT* construct was also assembled into pICH47811 (Addgene #48007) as the level 1 part, and then cloned at position 1 of the same binary vector in addition to the MAGT construct. The vector maps for these constructs are shown in Supplementary Fig. 6 and 7, respectively. All primers and DNA parts for Golden Gate cloning used in the present study are listed in Supplementary Data 3 and 4.

## Preparation of acceptor substrates

The alcohol-insoluble residue (AIR) of pine wood and the bottom part of the Arabidopsis stem were prepared by ball milling in ethanol followed by a successive wash with methanol:chloroform = 2:3 (v/v) and 65, 80, 100% ethanol.[20,37] For the preparation of seed mucilage AIR,[6] two grams of Arabidopsis seeds were soaked in 20 mL of water and were gently rotated at room temperature for 2 h. To release an adherent mucilage layer, the suspension was then shaken with an MM400 mixer mill (Retch) without beads at 30 Hz for 15 min twice. After centrifugation of the suspension, the supernatant was lyophilised. The dried seed mucilage was then washed with 70% ethanol twice to remove small sugars and dried at 45 °C. To extract pine AcGGM and Arabidopsis seed mucilage β-GGM, twenty mg of AIR were shaken at 1,000 rpm in 1 mL of 4 M potassium hydroxide solution at room temperature for 1 h. After centrifugation at 15,000 ×*g* for 15 min, the supernatant was collected, and the resulting pellet was washed with 1 mL of water. After centrifugation, both supernatants were combined, and the extracts were subjected to a PD-10 column (Cat no. GE17-0851-01, Cytiva) to exchange the solvent for 50 mM ammonium acetate buffer (pH 6.0). Ivory nut mannan was also treated the same. To extract the acetylated galactoglucomannan, forty mg of AIR was shaken at 1,000 rpm in 1 mL of water at 80 °C for 4 h. After centrifugation, the supernatant was collected. This was repeated three times to maximise the yield. The pooled supernatant was subjected to the PD−10 column to replace the solvent with the buffer. Those galactoglucomannans were treated with *Cellvibrio mixtus* GH27 α-galactosidase to remove innate Gal residues. The fractions corresponding to two mg AIR and 100 μg of ivory nut mannan were dried and used for the MAGT reaction. As oligosaccharide substrates, 1 mM mannohexaose was derivatised with fluorophore by incubation with a labelling reagent {0.2 M 8-aminonaphthalene−1,3,6-triflufonic acid (ANTS; Cat no. A350, Thermo Fisher Scientific), 0.2 M 2-picoline borane (Cat no. 654213, Sigma-Aldrich) in acetic acid:DMSO:$H_2O$ = 3:20:17} at 37 °C overnight. After

drying samples *in vacuo*, they were resuspended in water and 2 nmol of oligosaccharides were applied to the reaction.

## Transient protein expression in *N. benthamiana* and in vitro activity assay

The expression constructs were introduced into AGL-1 *Agrobacterium tumefaciens*, and the transformed Agrobacterium was then infiltrated into 5-week-old *N. benthamiana* leaves according to a published protocol.[61] For preparation of the membrane fraction,[62] the leaves were harvested five days after the infiltration, homogenised in a homogenisation buffer containing 50 mM HEPES-KOH (pH 7.0), 400 mM sucrose, 1 mM phenylmethanesulfonyl fluoride, 3% (w/v) polyvinylpolypyrrolidone, and cOmplete™ EDTA-free Protease Inhibitor Cocktail (Cat no. 11873580001, Roche). The resulting suspension was filtered through Miracloth (Cat no. 475855-1 R, Merck). The filtrate was centrifuged at 1,000 x*g* for 10 min and the supernatant was further centrifuged at 100,000 x*g* for 1 h. The resulting pellet was resuspended completely in a minimum amount of the homogenisation buffer without polyvinylpolypyrrolidone and stored at −80 °C until use. Protein concentration in the membrane fraction was determined by Bradford reagent (Cat no. B6916, Merck), and 5 μg of total proteins were subjected to SDS-PAGE. The proteins on the gel were blotted onto the nitrocellulose membrane using an iBlot system (Thermo Fisher Scientific). The MAGT proteins were immunologically probed by anti-Myc antibody (1:2,000 dilution in 1% milk, rabbit polyclonal, Lot no. 1007474-2, Cat no. ab9106, Abcam) as a primary antibody and goat anti-rabbit IgG HRP conjugate (1:10,000 dilution in 1% milk, Batch no. 64371828, Cat no. 1706515, Bio-Rad) as a secondary antibody. Amersham ECL Prime Western Blotting Detection Reagent (Cat no. RPN2232, Cytiva) was applied onto the membrane and chemiluminescence was scanned by ChemiDoc MP (Bio-Rad) with an auto-exposure setting. A pre-stained protein ladder was detected by colourimetric mode and the composite image with chemiluminescence detection was taken.

For the in vitro assay of MAGT activity,[20] the membrane fraction was mixed with a reaction mixture composed of 5 mM UDP-Gal (Cat no. 670111, Merck), 5 mM $MnCl_2$, 5 mM $MgCl_2$, 1 mM DTT, 0.1% Triton X-100, and desired acceptor substrates in a total volume of 60 μL. The reaction was done at room temperature for 4 h. To remove proteins and lipids, the reaction mixture was purified by the phase separation with methanol and chloroform.[63] For the reaction with pre-labelled oligosaccharides, the aqueous phase was dried *in vacuo* and analysed by PACE. For the reaction with polysaccharides, the products in the aqueous phase were further precipitated by ethanol and extensively washed with ethanol three times. The resulting pellet was dissolved in 50 mM ammonium acetate (pH 6.0) and treated with an excess of CjMan26A (1 unit), GH3 β-glucosidase (2 μg), and GH5 β-mannosidase (0.8 μg), in order to eliminate non-galactosylated products. The final products were labelled with ANTS by reductive amination.[64] The dried derivatised oligosaccharides were dissolved in 6 M urea and run by PACE in 0.1 M Tris-borate buffer (pH 8.2) at 200 V for 30 min and 1000 V for 2 h.[64] ANTS-labelled Man and mannnooligosaccharides with a degree of polymerisation of 2-6 were used as molecular standards. A G-box (Syngene) equipped with a transilluminator (wavelength 365 nm), and a detection filter (530 nm) were used to visualise the products in the gel.

## Transformation of Arabidopsis

The gene constructs were introduced into Arabidopsis by agrobacterium-mediated floral dipping.[65] Transformed seeds were screened by OLEO-GFP fluorescence under a fluorescent stereomicroscope, and each $T_1$ seed was numbered as an independent line and grown independently. Two or three homozygous lines of $T_3$ generation were used in all experiments.

## Observation and measurement of seed mucilage

Mature seeds were soaked in water and gently shaken at room temperature for 2 h. After water removal, seed mucilage was stained by 0.01% (w/v) ruthenium red for 10 min and observed by a light microscope. To measure the mucilage area, the seed and mucilage in the images were manually selected by Image J software. The area of mucilage was measured by subtracting the area of the seed from the whole area of mucilage.

## Monosaccharides composition analysis and cell wall fractionation

The bottom part of the inflorescence stem was harvested from 8-week-old plants (there was no obvious difference in growth phenotype between wild-type and transgenic plants). Plant material from six to nine plants was dealt as a set sample and three sets were grown independently for three biological replicates. AIR preparation was performed as described above. For monosaccharide composition analysis, a two-step acid hydrolysis was performed. Briefly, one mg of AIR was first hydrolysed with 200 µL of 72% sulphuric acid at 4 °C for 1 h. Then, 1.6 mL of water was added to the suspension and incubated at 100 °C for 4 h for the second hydrolysis step. After centrifugation at 15,000 × g for 10 min, the supernatant was neutralised with barium carbonate powder, then mixed with Dowex 50WX8 beads ($H^+$ form, Cat no. 217492, Merck). After centrifugation, the supernatant was collected and dried under the vacuum. The dried sample was dissolved in 100 µM inositol solution and analysed by high-performance anion-exchange chromatography/pulsed amperometric detection (HPAEC-PAD) using an ICS-3000+ fitted with a pulsed amperometric detector (Thermo Fisher Scientific). Monosaccharides in hydrolysates were separated by a CarboPac PA-20 column (3 × 150 mm; Thermo Fisher Scientific).[66] Data were acquired and analysed using Chromeleon software (version v6.80).

For cell wall fractionation, 20 mg of AIR were suspended in 1 mL of water and incubated at 80 °C for 5 h while shaken at 1000 rpm. After centrifugation at 15,000 ×g for 10 min, the supernatant was collected, and the resulting pellet was washed with 1 mL of water. After centrifugation, both supernatants were combined as the hot water (HW) fraction. The pellet was then treated with 50 mM ammonium oxalate at 25 °C for 1 h, and the supernatant was collected as the ammonium oxalate fraction. The resulting pellet was further treated with 4 M KOH as described above to extract the KOH fraction. For the monosaccharide composition of each fraction, samples were hydrolysed with 2 M trifluoroacetic acid (TFA) at 120 °C for 1 h. After drying under the vacuum, samples were dissolved in 100 µM of inositol solution and analysed by HAPEC-PAD as described above.

## Mannanase digestion, PACE, and oligosaccharides extraction from gels

KOH fractions extracted from one mg AIR were treated with an excess of either enzyme, CjMan26A (1 unit) or AnGH5 (3.5 µg) at 37 °C overnight. The mannanase products were further digested by various glycoside hydrolases, such as *Cellvibrio mixtus* GH27 α-galactosidase (1 µg), *Aspergillus niger* GH3 β-glucosidase (2 µg), and *Cellvibrio mixtus* GH5 β-mannosidase (0.8 µg) to analyse the structure of the mannanase products. The condition was the same except for reaction time, which was overnight for β-mannanase and α-galactosidase digestion or 4 h for β-glucosidase and β-mannosidase digestion. After each reaction, the enzyme was denatured by heating at 100 °C for 15 min. The final products were dried *in vacuo*, then derivatised with ANTS and subjected to PACE as described above.

To analyse the detailed structure, five milligrams of AIR were digested by CjMan26A, β-glucosidase, and β-mannosidase, prior to a preparative PACE using all lanes. Each band was excised from the gel on a transilluminator. The excised gels were ground in 500 µL of water in a microtube by a pestle and were shaken for 1 h in the dark. The

suspension was passed through a gravity-flow column and the ground gel was washed with 500 µL of water. Both eluents were combined and dried. The sample was reconstituted in 300 µL of water and the solvent was exchanged for 50 mM ammonium acetate (pH 6.0) by a PD mini-trap G-10 column (Cat no. 28918010, Cytiva). Sequential digestion was performed as described above. Dried samples were resolved in 10 uL of 6 M urea and analysed by PACE.

## Cellulose binding assay

For the in vitro cellulose binding assay,[67,68] Avicel PH-101 (Cat no. 11363, Sigma) as a cellulose source was resuspended in water at the concentration of 10 mg/mL. AcGGM was extracted with 4 M KOH from 50 mg AIR of Arabidopsis inflorescence stems as described above and was used after centrifugation at 15,000 xg for 5 min on the same day of the extraction to avoid precipitation. A mixture in a total volume of 1 mL containing 0.1 mg Avicel, 20 mM ammonium acetate pH 5.8, and the hemicellulose fraction corresponding to 0.05, 0.1, 0.2, 0.5, 0.75, 1, 2, 3 mg of AIR was incubated at 40 °C for 5 h shaking at 1,000 rpm for 1 min with 1 min intervals. The mixture without Avicel was also prepared for the control. After the incubation, the mixture was centrifuged at 15,000 ×g for 5 min and the supernatant was collected. Polysaccharides in the supernatant were hydrolysed by 2 N TFA as described above and the amount of mannose was measured by HPAEC-PAD. The amount of bound Man was determined by subtracting the amount of Man in solution with Aivcel from the total Man in the reaction without Avicel. To draw the adsorption isotherm, the amount of bound Man per mg Avicel ($B_e$) versus the concentration of free Man remaining in the solution at equilibrium ($C_e$) was plotted. The best-fit curve was obtained by a non-linear regression using GraphPad Prism 10 software (version 10.1.2), and $B_{max}$ and $K_d$ were calculated based on Eq. 1.

$$B_e = \frac{B_{max} \times C_e}{K_d + C_e} \tag{1}$$

Experiments were done with three technical replicates and repeated for three biological replicates that were independently grown and harvested.

## Solid-state NMR

Solid-state MAS NMR experiments were performed using a Bruker (Karlsruhe, Germany) AVANCE NEO ssNMR spectrometer, operating at $^1$H and $^{13}$C Larmor frequencies of 850.2 and 213.8 MHz, respectively, with 3.2 mm double-resonance E$^{free}$ MAS probe. Experiments were conducted at an indicated temperature of 283 K at a MAS frequency of 12.5 kHz. The $^{13}$C chemical shift was determined using the carbonyl peak at 177.8 ppm of L-alanine as an external reference with respect to tetramethylsilane. Both $^1$H–$^{13}$C CP, with ramped (70%–100%) $^1$H rf amplitude and 1 ms contact time, and DP were used to obtain the initial transverse magnetisation.[69] While CP emphasizes the more rigid material a short, 2 s, recycle delay DP experiment was used to preferentially detect the mobile components. Two-dimensional double-quantum correlation spectra were recorded using the refocused INADEQUATE pulse sequence which relies upon the use of isotropic, scalar J coupling to obtain through-bond information regarding directly coupled nuclei.[70–72] The carbon 90° and 180° pulse lengths were 3.5–4.3 and 7.0–8.6 µs, respectively, with 2τ spin-echo evolution times for a (π−τ−π/2) spin-echo of 4.48 ms. SPINAL-64 $^1$H decoupling was applied during both the evolution and signal acquisition periods at a $^1$H nutation frequency of 70–80 kHz.[73] The acquisition time in the indirect dimension ($t_1$) was 4.8-5.9 ms for the CP-INADEQUATE and 4.1-4.3 ms for the DP INADEQUATE experiment. The spectral width in the indirect dimension was 50 kHz for both with 64-96 acquisitions per $t_1$ FID for the CP-INADEQUATE and DP INADEQUATE experiments. The States-TPPI method was used to achieve sign discrimination in $F_1$. The

recycle delay was 2 s for both CP INADEQUATE and DP INADEQUATE experiments. The spectra were obtained by Fourier transformation into 4 K ($F_2$) x 2 K ($F_1$) points with exponential line broadening in $F_2$ of 70 Hz for CP and 60 Hz for DP experiments, respectively, and squared sine bell processing in $F_1$. All spectra obtained were processed and analysed using Bruker Topspin version 4.1.3.

## Statistics

Significant differences in the dataset presented here were determined by one-way ANOVA (two-tailed) followed by post hoc analysis using the Dunnet method compared with wild-type. Statistical values were shown in Supplementary Data 5 and 6. All statistics were run on GraphPad Prism 10 software (version 10.1.2).

## Reporting summary

Further information on research design is available in the Nature Portfolio Reporting Summary linked to this article.

## Data availability

All protein sequences used in this work were provided in Supplementary Data 2. The unprocessed solid-state NMR data are available in BMRB database under the entry ID BMRbig111 [https://bmrbig.org/released/bmrbig111]. The crystal structure of AtXXT1 is available in the PDB database under the PDB code 6BSW. Information on the magt1 Arabidopsis mutant are available from The Arabidopsis Information Resource (TAIR) under the code SALK_061576 [https://www.arabidopsis.org/germplasm?key=4664164]. The amino acid sequences of AtMAGT1 and AtMAGT2 are available in TAIR database under the accession codes At2g22900 [https://bioinformatics.psb.ugent.be/plaza/versions/plaza_v5_dicots/genes/view/AT2G22900] and At4g37690 [https://bioinformatics.psb.ugent.be/plaza/versions/plaza_v5_dicots/genes/view/AT4G37690], respectively. The amino acid sequences of PtMAGT and CtMAGT are available in NCBI database under the accession codes DAA64590.1 [https://www.ncbi.nlm.nih.gov/protein/DAA64590.1] and Q564G7.1 [https://www.ncbi.nlm.nih.gov/protein/Q564G7.1], respectively. Source data are provided with this paper.

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

## Acknowledgements

The authors would like to acknowledge Prof. George Lomonossoff (John Innes Centre, UK), who developed the pEAQ-HyperTrans expression system used in this study. Plant Bioscience Limited supplied the pEAQ-HT vector that was used in this work. AnGH5 mannanase and AnGH3 β-glucosidase were kindly provided by Novozymes A/S, Denmark. CmGH5 β-mannosidase was a kind gift from Harry Gilbert (University of New-castle). We wish to thank the Facility Manager Team (Dr Dinu Iuga and Dr Trent Franks, University of Warwick) for their help. This work was supported by Broodbank Research Fellowship to YY (reference no. PD16178), Herchel Smith PhD scholarship by the University of Cambridge to AEP, MEXT KAKENHI Grant-in-Aid for Scientific Research to TK (no. 23H02134 and no. 23H04302), and the ERC Advanced Grant EVOCATE to PD funded by the United Kingdom Research and Innovation (UKRI) grant number EP/X027120/1 (www.ukri.org) and by the grant OpenPlant (UKRI BBSRC, BB/L014130/1) to PD. The NMR at Warwick was supported by a Novo Nordisk Foundation grant Oxymist (Grant no. NNF20OC0059697). The UK High-Field ssNMR Facility used in this research was funded by EPSRC and BBSRC (EP/T015063/1), as well as the University of Warwick including via part funding through Birmingham Science City Advanced Materials Projects 1 and 2 supported by Advantage West Midlands (AWM) and the European Regional Development Fund (ERDF).

## Author contributions

YY and PD led the work. YY performed most of the experiments. JJL contributed to generating DNA constructs. XG and LY performed some in vitro assays, complementation of Arabidopsis mutants, and PACE of seed mucilage. AEP grew plants in a $^{13}$C chamber for solid-state NMR. RC, and RD performed solid-state NMR and YY, RC, and RD analysed data. TK and LY contributed to data interpretation. YY and PD wrote the manuscript, and all authors made comments on the manuscript.

## Competing interests

The authors declare no competing interests.
