## [Peer Review file · Nature Communications]

Glucomannan engineering highlights roles of galactosyl modification in fine-tuning cellulose-glucomannan interaction in Arabidopsis cell walls

Corresponding Author: Professor Paul Dupree

Version 0:

Reviewer comments:

Reviewer #1

(Remarks to the Author)

"Glucomannan engineering highlights the role of galactosyl modification in fine-tuning cellulose-glucomannan interaction in Arabidopsis cell walls" is a very thorough investigation of mannan/glucomannan functional properties. The subtle but very significant substrate specificity differences between α -galactosyltransferases are characterised in detail (and make sense). Also, the effects of patterning of side-chains on the glucomannan backbone on interactions with cellulose are clearly presented.

It is a very interesting and significant paper and I actually do not have any concerns or reservations as to the technical quality or completeness of the work. However, I want to encourage the authors to consider a few stylistic things:

The discussion is ambitious - it wants to do so many things that distillation and recrystallisation of the key take home messages are lost to some extent. A bit shorter and, if permitted by the journal, with some sub-heads thrown in to make the structure clear would do good. The saved space could be used for a bit more hand-holding in the Results section. I would benefit from some help on p.9. It is not the existence or absence of Glc in the backbone which is unique to Gymnosperms (as we are reminded of on the following page) but rather the repeating disaccharide which is important. What is it that is shared between glucomannan fine structure in ferns and gymnosperms that correlates with the key Leu?

Finally, I think that Dhugga et al (DOI: 10.1126/science.1090908) deserve a citation for the discovery of the guar mannan synthase.

Reviewer #2

(Remarks to the Author)

In this study, the authors investigate the substrate specificity of four galactosyltransferases that function in adding side galactose to mannose in mannan or glucomannan backbone. They selected two Arabidopsis MAGTs that were annotated in family GT34, and two others were selected from pine and guar beans due to differences in galactomannan pattern in their cell walls. Using heterologous expression in *N. benthamiana* leaves, complementary studies in Arabidopsis, combined with biochemical characterization of the products synthesized and their patterns and with bioinformatic structural modeling, the authors concluded that different isoforms of MAGTs are responsible for different patterns of synthesized mannans, which is, to my opinion, is not really anything new and quite logical. Plants carry large families of GTs with a similar type of donor substrate specificity which should already suggest that they should recognize different acceptors and synthesize polysaccharides with diverse patterns observed in cell walls. Examples from other GTs characterized so far support this. In addition to modeling and product characterization, the authors also investigated the impact of differently branched galactoglucomannans on their interaction with cellulose, which could impact the cell wall properties and extractability of mannans.

Main comments:

The crystal structure of Arabidopsis XXT1 was used to model the structure of MAGTs and to explain their acceptor substrate specificity and how they are able to synthesize specific patterns of backbone galactosylation. What was the level of confidence in the prediction of the folding using XXT1? This information has to be added to convince that all discussions based on comparison with XXT1 are valid. It would be more convincing if they would generate the mutants, swapping L278

residues in PtMAGT. In supplementary Figure 4, I did not find panel f, although there is a description in the legend related to that panel. I wonder, since binding sites of MAGTs were compared with XXT1, what else is different that determines their specificity to mannoses in the mannan backbone and not glucose in the glucan backbone, which is in xyloglucan? What could determine their involvement in glucomannan/mannan galactosylation and not glucan?

In discussion, the speculation about the isolation of AtGGM and β -GGM substrates for MAGTs in time or space is very speculative. The authors do not have any evidence for that.

Minor:

On lane 255, "...MAGT can transfer different patterns", needs revision. Enzymes cannot transfer the patterns they catalyze the reaction sugar transfer.

On lanes 258-260, what would be an alternative to this conclusion? Otherwise, as I pointed out above, this statement is quite obvious, taking into consideration other studies on GTs, including those that are involved in secondary metabolite glycosylation.

On lanes 297-299, Small amounts of galactosylated products produced in Arabidopsis do not give confidence in potential applicability. I am sure the availability of acceptor substrates would be quite critical for such applications.

In the legend for Figure 6, add the meaning of circles and Xs.

In suppl figure 4, the panel f is missing.

Reviewer #3

(Remarks to the Author)

In this study, Yoshimi and coworkers investigated the role of galactosyl modification on Arabidopsis cell walls. This included that they used solid-state MAS NMR and biochemical assays to probe the effect of galactosylation. The results suggested that Gal promotes AcGGM binding to cellulose. However, excess of Gal decreases AcGGM interactions.

The solid-state MAS NMR experiment, 2D INADEQUATE, was performed on stems. Additionally, both CP and DP were used to identify polymers with different hydrodynamical states.

For the wild-type stems, cellulose, which is rigid, contributed most signal to the CP-refocused INADEQUATE spectra. However, AcGGM-related peaks were also observed in the CP spectrum. In contrast, while peaks were observed in the DP spectrum (collected with a short recycle delay) that correlated to the mobile species, pectic galactan and arabinan, there were not any peaks due to AcGGM. This indicated that AcGGM in wild-type stems is rigid. For the pIRX3::AtMAGT2 stems, peaks due to AcGGM were observed in the DP-refocused spectrum, and observed to a lesser extent in the CP spectrum, indicating that these species are somewhat mobile in this strain background.

Overall, this was a very nice application of these NMR experiments to show changes in mobility of different glycan species upon galactosylation. The NMR methodology and analyses are sound, and I have no concerns or major comments, only a suggestion. Perhaps it would be interesting, albeit potentially challenging given the spectral overlap, to collect comparative DARR spectra of the wild-type and pIRX3::AtMAGT2 stems to monitor changes in AcGGM-cellulose interactions directly.

Version 1:

Reviewer comments:

Reviewer #1

(Remarks to the Author)

The authors have responded constructively and I have no further concerns

Reviewer #2

(Remarks to the Author)

most of my concerns were addressed

Reviewer #3

(Remarks to the Author)

My comments were addressed.

RESPONSE TO REVIEWER COMMENTS

Reviewer #1 (Remarks to the Author):

>>>"Glucomanan engineering highlights the role of galactosyl modification in fine-tuning cellulose-glucomanan interaction in Arabidopsis cell walls" is a very thorough investigation of mannan/glucomanan functional properties. The subtle but very significant substrate specificity differences between α -galactosyltransferases are characterised in detail (and make sense). Also, the effects of patterning of side-chains on the glucomanan backbone on interactions with cellulose are clearly presented.

It is a very interesting and significant paper and I actually do not have any concerns or reservations as to the technical quality or completeness of the work. However, I want to encourage the authors to consider a few stylistic things:

We thank the Reviewer 1 for helpful feedback. We have made changes in the manuscript based on their comments and a point-by-point response is below.

>>The discussion is ambitious - it wants to do so many things that distillation and recrystallisation of the key take home messages are lost to some extent.

Thank you for suggestion. We went through the whole discussion and we shorten some parts of discussion and added some sentences to highlight key messages from our work.

>> A bit shorter and, if permitted by the journal, with some sub-heads thrown in to make the structure clear would do good.

As subheadings in Discussion is not recommended according to the journal's guidelines, we decided not to have them.

>>The saved space could be used for a bit more hand-holding in the Results section. I would benefit from some help on p.9.

We revised some parts of result section for better clarity.

>>It is not the existence or absence of Glc in the backbone which is unique to Gymnosperms (as we are reminded of on the following page) but rather the repeating disaccharide which is important. What is it that is shared between glucomanan fine structure in ferns and gymnosperms that correlates with the key Leu?

As far as we know, the Glc-Man repeating structure are unique to β -GGM that exist exclusively in eudicots. According to the literature, the structure of AcGGM in gymnosperms, and ferns are fairly similar to each other, where they all have random arrangements of Glc and Man in the backbone (probably with a minor difference in the ratio of Man:Glc:Gal). Unfortunately, however, there is no information on the detailed structure of AcGGM in ferns, specifically where galactosylation occurs on the backbone. Moreover, most GT34s (except for one in *Salvinia cucullata*) from ferns do not belong to the MAGT clade in the GT34 family phylogenetic tree. We checked the residues of all GT34s in the phylogenetic tree and actually the fern GT34s possess either a non-polar amino acid or threonine at subsite 3. Probably, it is too early to conclude at this point that it is conserved in early land plants without knowing their actual activities. Therefore, we rephrased the text in the result section (page 10, line 199-200) and discussion (page 18-19, line 396-402).

>>Finally, I think that Dhugga et al (DOI: 10.1126/science.1090908) deserve a citation for the discovery of the guar mannan synthase.

Yes, we recognise the work by Dhugga et al identifying the mannan biosynthetic enzyme in plants for the first time and we cited this in the introduction as the reference 15 (page 5,

line 98).

Reviewer #2 (Remarks to the Author):

>>>In this study, the authors investigate the substrate specificity of four galactosyltransferases that function in adding side galactose to mannose in mannan or glucomannan backbone. They selected two Arabidopsis MAGTs that were annotated in family GT34, and two others were selected from pine and guar beans due to differences in galactomannan pattern in their cell walls. Using heterologous expression in *N. benthamiana* leaves, complementary studies in Arabidopsis, combined with biochemical characterization of the products synthesized and their patterns and with bioinformatic structural modeling, the authors concluded that different isoforms of MAGTs are responsible for different patterns of synthesized mannans, which is, to my opinion, is not really anything new and quite logical. Plants carry large families of GTs with a similar type of donor substrate specificity which should already suggest that they should recognize different acceptors and synthesize polysaccharides with diverse patterns observed in cell walls. Examples from other GTs characterized so far support this. In addition to modeling and product characterization, the authors also investigated the impact of differently branched galactoglucomannans on their interaction with cellulose, which could impact the cell wall properties and extractability of mannans.

Whilst it is true that it is well known different GTs have different specificities, this has not been explored in MAGTs. The only previous work on AtMAGT1 showed a requirement for Glc in the backbone, but did not explore this further (Yu et al., 2018). Our manuscript reveals this specificity clearly for multiple MAGTs, and exploits this for wall engineering.

Main comments:

>>>The crystal structure of Arabidopsis XXT1 was used to model the structure of MAGTs and to explain their acceptor substrate specificity and how they are able to synthesize specific patterns of backbone galactosylation. What was the level of confidence in the prediction of the folding using XXT1? This information has to be added to convince that all discussions based on comparison with XXT1 are valid.

Thank you for the important suggestion. Yes, it is crucial for discussion that the predicted models have high confidence level. In fact, our predicted structures had a high pIDDT (over 94) score, meaning a very high confidence level, and this was also the case for individual amino acid residues that we discussed in the text. We added a sentence about this (page 9, line 183-184) and a supplementary figure to show the confidence level of the model structure to support our discussion.

>>>It would be more convincing if they would generate the mutants, swapping L278 residues in PtMAGT.

Thank you for your suggestions. This could be interesting but is a set of experiments that would take many months to generate and analyse the plants. The scope of this manuscript is to reveal the importance of the specificity in determining galactosylation patterns and the importance of the patterns for wall assembly, and we speculate about how such enzyme specificity might arise. To show how the enzymes generate the specificity entails substantial experiments, and in the future could be of interest.

>>>In supplementary Figure 4, I did not find panel f, although there is a description in the legend related to that panel.

We apologise for the mistake. The panel f in the legend referred to the panel e in the supplementary Figure 4. We corrected the legend to match the figure.

>>>I wonder, since binding sites of MAGTs were compared with XXT1, what else is different that determines their specificity to mannoses in the mannan backbone and not glucose in the glucan backbone, which is in xyloglucan?
What could determine their involvement in glucomannan/mannan galactosylation and not glucan?

These are indeed good questions. From the comparison of MAGT model structures and XXT1 structure, we could not find any conclusive difference to explain why MAGT recognise Man to transfer Gal on and what determines MAGTs' activities toward mannan/glucomannan/glucan. For the first question, amino acid residues recognising orientation of C2 hydroxyl (axial or equatorial) at the subsite 4 could have a key role in differentiating Glc and Man. In XXT1 there is a tyrosine residue in a 'gating loop' over the subsite 4 recognising the C2 hydroxyl of Glc in glucan and this is conserved in MAGTs too. However, the 'gating loop' seems longer in MAGTs so there are differences that might be involved in acceptor recognition. In addition, difference in donor substrates between XXTs and MAGTs might also contribute to differentiating acceptor substrates through a conformational change upon donor binding. These are interesting themselves; however, our work focuses on discovering variable substrate specificities of MAGT that create different Gal modification patterns on glucomannan.

>>>In discussion, the speculation about the isolation of AtGGM and β -GGM substrates for MAGTs in time or space is very speculative. The authors do not have any evidence for that.

We showed that AtMAGT1 and AtMAGT2 are able to galactosylate glucomannan *in vitro* and *in vivo*, yet glucomannan in wild-type Arabidopsis has very little Gal. In the primary cell walls, both AcGGM and beta-GGM coexist (Yu et al., 2022) so if they were synthesised in the same pool of substrates, both could be similarly galactosylated unless there is a mechanism of making synthetic protein complexes or compartmentalisation (by Golgi or within Golgi). Although we don't have direct evidence supporting this idea, we believe that our data suggest that there is unknown mechanism controlling the galactosylation in the presence of both AcGGM and beta-GGM. We rephrased some sentences regarding this (page 20, line 426-432). We hope the revision sounds less assertive and makes our point clearer.

Minor:

>>>On line 255, "...MAGT can transfer different patterns", needs revision. Enzymes cannot transfer the patterns they catalyze the reaction sugar transfer.

We have corrected the sentence (page 12, line 258-259).

>>>On lines 258-260, what would be an alternative to this conclusion? Otherwise, as I pointed out above, this statement is quite obvious, taking into consideration other studies on GTs, including those that are involved in secondary metabolite glycosylation.

There were some hypotheses why different plant species have distinct galactosylation on cell wall mannans, for example, 1) the MAGTs have different activities (specifically that they recognise the acceptor differently so that they add different amount of Gal with a different patterning), or 2) all MAGTs in the plant kingdom produce the same galactosylation patterns regardless of the acceptor substrates but some of them are removed by galactosidase post-biosynthetically. Our data showed that the former is the case although we cannot totally exclude whether the post-synthetic modification also exists. It might be an obvious hypothesis to some that MAGTs differ in specificity, but to our knowledge, there has not been any work addressing how the galactosylation on glucomannan is determined in nature. In our work, we systematically addressed this and found that different MAGTs modified glucomannan differently.

As the Reviewer 2 mentioned, plants carry a large number of families of GTs responsible for biosynthesis of plant cell wall polysaccharides; however, surprisingly, there are only a few examples of different GTs within the same clade of a GT family that share the same donor but produce different structures of polysaccharides. For example, GUX1 vs GUX2 in GT8 for GlcA patterning of xylan (Bromley et al., 2013), CslA2 vs CslA9 in GT2 for different ratio of Man/Glc in glucomannan backbones (Yu et al., 2022), and MUR3/XLT2 vs MBGT in GT47 for xyloglucan or β GGM β -1,2-galactosylation (Yu et al., 2022; Ishida et al 2024). We believe that our work uncovers one important new aspect concerning how galactosylation of glucomannan is controlled.

>>>On lanes 297-299, Small amounts of galactosylated products produced in Arabidopsis do not give confidence in potential applicability. I am sure the availability of acceptor substrates would be quite critical for such applications.

Here in our work we showed the feasibility of glucomannan engineering in planta even if the acceptor substrate in Arabidopsis is limited. The amount of galactosylation and the subsequent effects would be more enhanced if we could use this technique in a different system where more glucomannan substrates are available for MAGTs, perhaps in gymnosperms for example. Further, the availability of acceptor can be engineered in plants. We believe that our work presents a new method of glucomannan engineering that could be used for many applications. We add a sentence in discussion regarding this point (page 24, line 508-510).

>>>In the legend for Figure 6, add the meaning of circles and Xs.

Thank you for pointing it out. We added sentences to explain the circles and crosses.

>>>In suppl figure 4, the panel f is missing.

We apologise for the mistake. As mentioned above, the legend of Supplementary Fig. 4 is now corrected.

Reviewer #3 (Remarks to the Author):

>>>In this study, Yoshimi and coworkers investigated the role of galactosyl modification on Arabidopsis cell walls. This included that they used solid-state MAS NMR and biochemical assays to probe the effect of galactosylation. The results suggested that Gal promotes AcGGM binding to cellulose. However, excess of Gal decreases AcGGM interactions. The solid-state MAS NMR experiment, 2D INADEQUATE, was performed on stems. Additionally, both CP and DP were used to identify polymers with different hydrodynamical states.

For the wild-type stems, cellulose, which is rigid, contributed most signal to the CP-refocused INADEQUATE spectra. However, AcGGM-related peaks were also observed in the CP spectrum. In contrast, while peaks were observed in the DP spectrum (collected with a short recycle delay) that correlated to the mobile species, pectic galactan and arabinan, there were not any peaks due to AcGGM. This indicated that AcGGM in wild-type stems is rigid. For the pLRX3::AtMAGT2 stems, peaks due to AcGGM were observed in the DP-refocused spectrum, and observed to a lesser extent in the CP spectrum, indicating that these species are somewhat mobile in this strain background.

Overall, this was a very nice application of these NMR experiments to show changes in mobility of different glycan species upon galactosylation. The NMR methodology and analyses are sound, and I have no concerns or major comments, only a suggestion. Perhaps it would be interesting, albeit potentially challenging given the spectral overlap, to collect comparative DARR spectra of the wild-type and pLRX3::AtMAGT2 stems to monitor changes in AcGGM-cellulose interactions directly.

Thank you for the suggestion. As the reviewer 3 mentioned, the spectral overlap as well as the fact that glucomannan in Arabidopsis is low abundance, it was hard to find the cross peaks between AcGGM and cellulose to show their interaction directly. For this, a glucomannan-rich system (such as gymnosperm etc.) will be required, and so we believe the experiment proposed is not yet feasible.

REVIEWERS' COMMENTS

Reviewer #1 (Remarks to the Author):

>*The authors have responded constructively and I have no further concerns*

We appreciate the reviewer's constructive comments and suggestions.

Reviewer #2 (Remarks to the Author):

>*most of my concerns were addressed*

We sincerely appreciate the reviewer's insightful comments.

Reviewer #3 (Remarks to the Author):

>*My comments were addressed.*

We are grateful for the reviewer's valuable comments and helpful suggestions.